# Oculomotor inhibition precedes temporally expected auditory targets

Dekel Abeles[1,4], Roy Amit[2,4], Noam Tal-Perry [1], Marisa Carrasco[3] & Shlomit Yuval-Greenberg [1,2✉]

Eye movements are inhibited prior to the onset of temporally-predictable visual targets. This oculomotor inhibition effect could be considered a marker for the formation of temporal expectations and the allocation of temporal attention in the visual domain. Here we show that eye movements are also inhibited before predictable auditory targets. In two experiments, we manipulate the period between a cue and an auditory target to be either predictable or unpredictable. The findings show that although there is no perceptual gain from avoiding gaze-shifts in this procedure, saccades and blinks are inhibited prior to predictable relative to unpredictable auditory targets. These findings show that oculomotor inhibition occurs prior to auditory targets. This link between auditory expectation and oculomotor behavior reveals a multimodal perception action coupling, which has a central role in temporal expectations.

[1] School of Psychological Sciences, Tel-Aviv University, Ramat Aviv, 6997801 Tel Aviv-Yafo, Israel. [2] Sagol School of Neuroscience, Tel-Aviv University, Ramat Aviv, 6997801 Tel Aviv-Yafo, Israel. [3] Department of Psychology and Center for Neural Science, New York University, 6 Washington Place, New York, NY 10003, USA. [4] The authors contributed equally: Dekel Abeles, Roy Amit. ✉email: shlomitgr@tau.ac.il

Temporal expectations are formed based on temporal regularities, and can be used to distribute processing resources effectively across time. The effect of temporal expectations on perceptual readiness is often demonstrated by enhanced behavioral performance, i.e., faster reaction times (RTs) and higher accuracy rates for anticipated targets[1]. However, these traditional behavioral correlates of temporal expectations provide only a retrospective evaluation of information processing, as they are assessed only after target onset, once the formation of expectations has already been completed. In contrast, monitoring eye movements can provide a reliable estimate of temporal expectations, while they are being formed, i.e., prior to the target appearance. We have found that saccades and blinks are more strongly inhibited prior to the appearance of a predictable, relative to an unpredictable, visual target. This pretarget oculomotor effect emerged with targets embedded in a rhythmic stream of stimulation[2], with targets associated with temporal cues[3], and in a temporal attention task in which the time of the target was fully predictable and selective attention was manipulated[4].

The purpose of this pretarget oculomotor inhibition is still unknown. Given that we had investigated this effect with visual targets only, we hypothesized that oculomotor inhibition could support vision by reducing the occurrence of eye movements and blinks during target presentation, which could impair target detection and discrimination.

The purpose of the present study is to examine whether pretarget oculomotor inhibition is evident also prior to predictable auditory targets. The question of whether pretarget oculomotor inhibition effect is present in nonvisual modalities has important implications for explaining this effect. Finding no oculomotor inhibition prior to predictable auditory targets would indicate that this effect reflects a within modality perception action coupling. Alternatively, finding an oculomotor inhibition effect for auditory targets would imply the existence of a multimodal perception action coupling.

Only a few studies have shown that nonvisual processes can modulate eye movements during or after stimulation. For example, in audition, microsaccades are inhibited following stimulus presentation[5,6] and their direction is biased towards the locus of auditory attention[7]. Furthermore, cognitive load modulates oculomotor activity in auditory tasks[8,9] and even in mental arithmetic tasks[10,11]. However, it is yet unknown whether eye movements are modulated prior to nonvisual tasks, i.e., whether they reflect nonvisual expectation.

In this study we investigate the relation between oculomotor inhibition and auditory temporal expectations. In two experiments, we manipulate temporal expectations using an auditory temporal cue, while jittering the intervals between trials to avoid a rhythmic stream of auditory stimuli (Fig. 1). Gaze positions were monitored while participants performed an auditory discrimination task preceded by temporally predictive or nonpredictive auditory cues. In Experiment 1, we manipulated the interval between the cue and the target, called foreperiod, to be either predictable or unpredictable. In the predictable blocks 100% of the trials are composed of the same foreperiod, whereas in the unpredictable blocks the foreperiods are chosen randomly out of five possible options per trial (1/1.5/2/2.5/3 s). Results reveal that saccades and blinks are inhibited prior to predictable auditory targets. In Experiment 2 we evaluate whether oculomotor inhibition is also modulated by probabilistic predictability, i.e., when targets are only partially predictable. This second experiment is similar to Experiment 1, except that the predictable blocks include 80% trials with one foreperiod (1 s) and 20% with another (2.2 s). In the unpredictable blocks of Experiment 2 the foreperiods are chosen randomly out of five possible options (1–3 s in 0.5 s steps, as in Experiment 1). Results of both

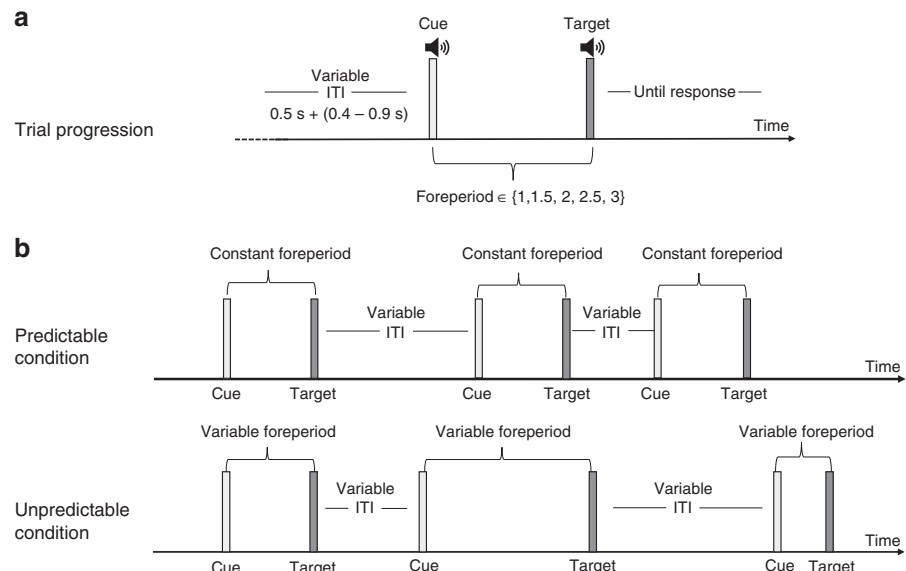

**Fig. 1 Experimental procedure of Experiment 1. a** After an online gaze contingent procedure confirmed fixation (<0.5° off center) and following an additional random inter-trial-interval (ITI; 0.4–0.9 s), the temporal cue (pure tone of 5 KHz) was played for 33 ms, marking the onset of the foreperiod (1/1.5/2/2.5/3 s). After the foreperiod, the target tone (descending or ascending chirp sound) was played for 33 ms and participants were asked to perform a 2-alternative forced choice (2AFC) discrimination task: report whether the chirp was ascending or descending by pressing one of two buttons. Participants were instructed to be as accurate as possible and to respond within the 4 seconds response window. Following the response, or after 4 s without one, the fixation cross changed color to gray for 200 ms to signal the end of the trial. **b** The foreperiod was either constant throughout the block (predictable condition) or changed randomly in different trials within the same block (unpredictable condition). Thus, the cue acted as a 100% valid temporal cue in the predictable condition but was uninformative regarding target timing in the unpredictable condition. The stimuli were identical in the two conditions, and differed only in the validity of the temporal cue in predicting the time of the target. Participants were not informed as to any predictability; therefore, all temporal expectations were learned incidentally.

experiments reveal that saccades and blinks are inhibited prior to predictable auditory targets. We conclude that pretarget oculomotor inhibition reflects multimodal perception action coupling, which could function as a mechanism of temporal expectation. Thus, future studies could use pretarget oculomotor inhibition effects as a biomarker of temporal expectation.

## Results

**Experiment 1**. Behavioral performance: accuracy-rates and reaction times: Accuracy-rates and reaction times (RTs) were calculated separately for each participant, condition and foreperiod. A two-way repeated-measures ANOVA with factors Predictability (predictable/unpredictable) and Foreperiod (1/1.5/ 2/2.5/3 s) revealed no evidence for differences in accuracy-rates between predictability conditions ($F_{(1,19)} = 1.62$, $p = 0.22$) or foreperiods ($F_{(4,76)} = 0.81$, $p = 0.52$), and no significant interaction between these two factors ($F_{(4,76)} = 0.39$, $p = 0.746$). The same analysis performed on RT of correct trials (secondary variable) revealed a significant main effect of foreperiod ($F_{(4,76)} = 4.83$, $p = 0.006$, $\epsilon = 0.708$, $\eta_p^2 = 0.203$), no significant main effect of Predictability ($F_{(1,19)} = 1.44$, $p = 0.24$), and no significant interaction of these two factors ($F_{(4,76)} = 1.87$, $p = 0.15$, $\epsilon = 0.669$). We conducted trend analysis across foreperiods separately for each predictability condition. A significant positive linear trend was evident in the predictable condition ($F_{(1,19)} = 6.815$, $p = 0.017$, $\eta_p^2 = 0.264$), as expected from the known relation between RT and foreperiod[3,12]. No significant trend was found in the unpredictable condition ($F_{(1,19)} = 1.808$, $p = 0.195$). These findings are depicted in Fig. 2.

**Saccades**. Pretarget saccade rate. The time series of saccade rate were constructed for each participant and condition and smoothed using a sliding window of 50 ms. A two-way repeated-measures ANOVA was performed on the average saccade rate at $-100$ to 0 ms relative to target onset, with factors Predictability (predictable/unpredictable) and Foreperiod (1/1.5/2/2.5/3 s). There was a significant effect of Predictability ($F_{(1,19)} = 21.943$, $p < 0.001$, $\eta_p^2 = 0.536$), arising from stronger inhibition of saccades in the predictable than the unpredictable condition. This predictability effect indicates that saccade-inhibition was a marker for the ability to anticipate the occurrence of an expected event (Fig. 3). This effect is consistent with our findings in the visual domain[2,3]. A significant main effect of foreperiod ($F_{(4,76)} = 3.241$, $p = 0.016$, $\eta_p^2 = 0.146$) indicated that saccade rate varied among foreperiods, but there were no significant linear or quadratic trends for this variation (Linear: $F = 0.131$, $p = 0.721$; Quadratic: $F = 1.778$, $p = 0.198$). The interaction between Predictability and Foreperiod was not significant ($F_{(4,76)} = 1.288$, $p = 0.282$), suggesting that the predictability effect of oculomotor inhibition was similarly evident in all foreperiods (Fig. 4a). We conducted separate trend analyses on each predictability condition and found no significant linear or quadratic trends in either of them.

Saccade rate slope. To examine the evolution of oculomotor inhibition over time, we calculated the slope of saccade rate across time as the difference between the average saccade rate at the pretarget window ($-100$ to 0 ms relative to target onset) and the average saccade rate at 400–500 ms post-cue, divided by the time between these two windows in seconds. A two-way repeated-measures ANOVA was conducted with saccade rate slope as the dependent variable and foreperiod and predictability as the independent variables (Fig. 4b). There was a significant main effect for predictability condition ($F_{(1,19)} = 5.08$, $p = 0.036$, $\eta_p^2 = 0.211$) resulting from a steeper slope in the predictable than the

unpredictable condition. There was a significant main effect of foreperiod ($F_{(4,76)} = 5.923$, $p = 0.012$, $\epsilon = 0.380$, $\eta_p^2 = 0.238$) indicative of a negative linear trend: the slope was shallower for longer foreperiods ($F_{(1,19)} = 14.482$, $p < 0.001$, $\eta_p^2 = 0.433$). The interaction between foreperiod and predictability was not significant ($F_{(4,76)} = 0.604$, $p = 0.661$, $\epsilon = 0.520$). We conducted separate trend analyses on the two predictability conditions. This analysis revealed a significant positive linear trend in the predictable condition ($F_{(1,19)} = 10.158$, $p = 0.005$, $\eta_p^2 = 0.348$), reflecting steeper slopes for shorter than for longer foreperiods, but only a marginal positive linear trend in the unpredictable condition ($F_{(1,19)} = 3.746$, $p = 0.068$). This may suggest that, consistently with our findings in the visual modality[3], the saccade rate slope was adjusted according to the expected foreperiod duration to reach maximal inhibition at target onset.

**Blink rate**. Pretarget blink rate. The time series of blink rate were constructed for each participant and condition and smoothed using a sliding window of 100 ms. A two-way repeated-measures ANOVA with factors Predictability and Foreperiod was performed on the average pretarget blink rate at $-500$ to 0 ms relative to target onset. This analysis revealed a main effect of Predictability ($F_{(1,19)} = 5.568$, $p = 0.029$, $\eta_p^2 = 0.227$; Fig. 5), a significant main effect of Foreperiod ($F_{(4,76)} = 4.555$, $p = 0.015$, $\epsilon = 0.537$, $\eta_p^2 = 0.193$) and a significant interaction between them ($F_{(4,76)} = 4.66$, $p = 0.008$, $\epsilon = 0.657$, $\eta_p^2 = 0.197$). In the predictable condition, a significant positive linear trend was found for foreperiod ($F_{(1,19)} = 6.09$, $p = 0.023$, $\eta_p^2 = 0.243$), reflecting an increase in blink rate with increased foreperiod duration. In contrast, in the unpredictable condition, no significant linear trend was found ($F_{(1,19)} = 0.18$, $p = 0.067$) but a significant positive quadratic trend ($F_{(1,19)} = 14.267$, $p < 0.001$, $\eta_p^2 = 0.429$) emerged. This suggests that in the unpredictable condition inhibition was strongest at the average foreperiod of 2 s and gradually increased towards shorter and longer foreperiods. Blink rate slopes analysis was not conducted as blinks were too sparse to reliably estimate the slope of their rate function.

**Behavioral consequences of oculomotor events**. We examined the perceptual consequences of oculomotor inhibition using two approaches: (1) We compared the behavioral measures of trials in which saccades overlapped with stimulus presentation (saccade onset at $-10$ to 33 ms relative to target onset), and trials in which no saccades were found in this interval; (2) We compared pretarget ($-100$ to 0 ms relative to target onset) saccade rate in correct vs. incorrect trials and fast vs. slow response trials, divided according to the median. For both of these analyses, we focused only on trials of the unpredictable condition in which any differences found between trials with and without saccades could be attributed to the influence of the saccades per se, rather than to the formation of cue-based temporal expectations. In contrast, in the predictable condition, it is impossible to dissociate whether effects would emerge because saccades may have interfered with auditory perception or because temporal expectations were not precise enough in these trials and thus saccades were not suppressed at the right timing.

(1) Paired sample t-tests were conducted separately for accuracy-rates and RTs of the unpredictable condition to compare performance in saccade trials (trials with saccades at $-10$ to 33 ms relative to target onset) and no saccade trials, collapsed across foreperiods. No significant differences were found between the two trial types in accuracy-rates (with saccades:

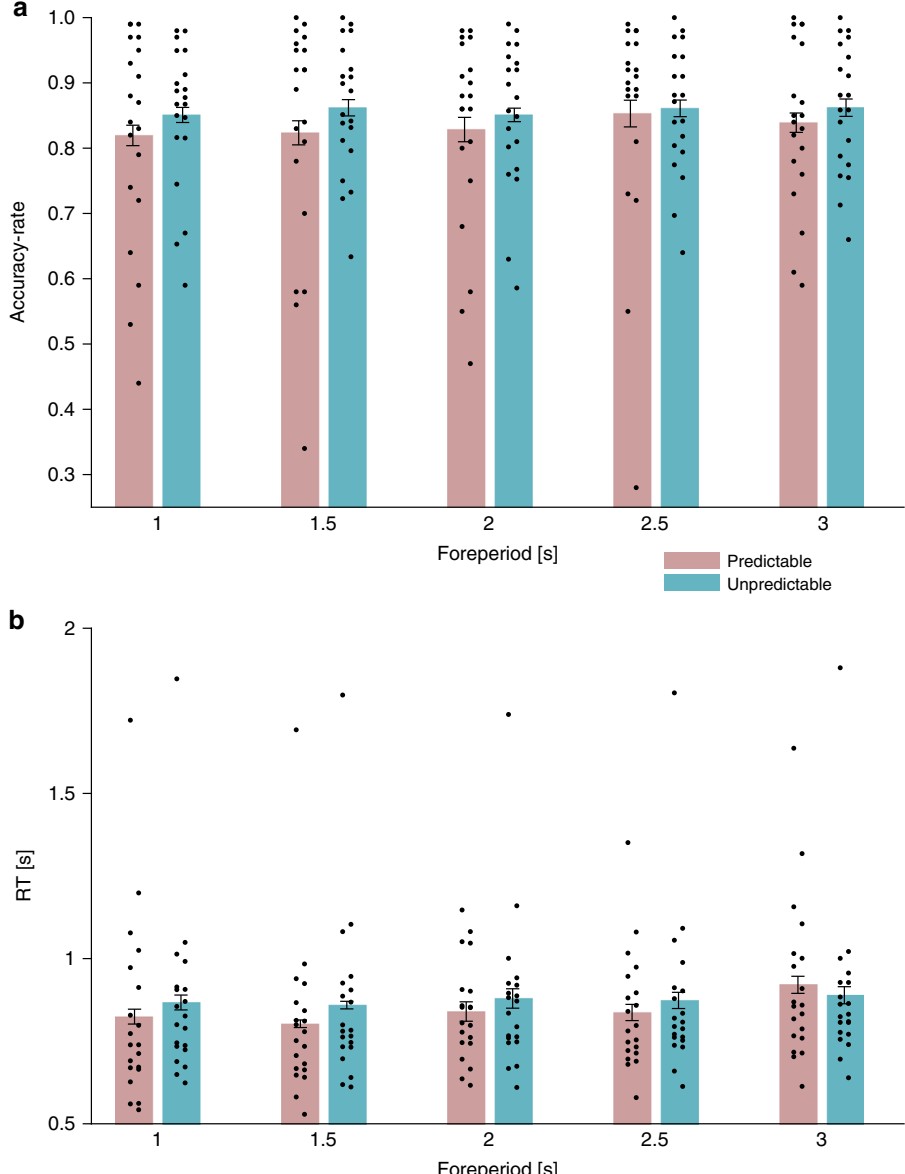

**Fig. 2 Accuracy-rates and reaction times (RTs) by predictability and foreperiod. a** Accuracy-rates in predictable (pink bars) and unpredictable (turquoise bars) conditions. $N = 20$ participants. **b** Reaction times in predictable (pink bars) and unpredictable (turquoise bars) conditions. Error bars denote ±1 standard error of the mean, corrected for within-subjects variability[60]. $N = 20$ participants. Source data are provided as a Source Data file.

mean = 0.867, SD = 0.11; without saccades: mean = 0.861, SD = 0.097; $t(19) = 0.59$, 95% CI = [−0.017 0.031], $p = 0.563$) and RTs (with saccades: mean = 0.927, SD = 0.2; without saccades: mean = 0.901, SD = 0.263; $t(19) = 1.28$, 95% CI = [−0.016 0.067], $p = 0.214$), suggesting that the occurrence of a saccade during target presentation did not influence performance in the task.

(2)  Two paired sample $t$-tests were conducted to compare the pretarget saccade rates in correct vs. incorrect trials and in slow vs. fast trials based on a median split. No significant differences in pretarget saccade rate were found between high and low performance trials (Correct: mean = 1.556, SD = 0.47; Incorrect: mean = 1.334, SD = 0.816: Correct vs. Incorrect: $t(19) = 1.40$, 95% CI = [−0.111 0.555], $p = 0.178$; Fast: mean = 1.561, SD = 0.524; Slow: mean = 1.476, SD = 0.482; Fast vs. Slow: $t(19) = 1.54$, 95% CI = [−0.03 0.201], $p = 0.139$, Fig. 6). Blinks occurrences during target

presentation were too rare to allow performing a similar analysis on blinks.

**Experiment 2.** In the predictable condition of Experiment 1 there was 100% certainty regarding the timing of the target relative to the cue. We found that saccades and blinks are inhibited prior to the 100% predictable targets. The purpose of this experiment was to examine whether this inhibition also occurs when predictability is probabilistic; i.e., for targets that are mostly, instead of fully, predictable. We hypothesized that saccades and blinks will be inhibited prior to the most probable target onset even in this condition in which the intervals are not constant.

In Experiment 1 we established the predictability effects across different foreperiods. In this second experiment, which required more trial repetitions than the previous one, we decided to focus solely on one foreperiod, and consequently avoid the necessity of

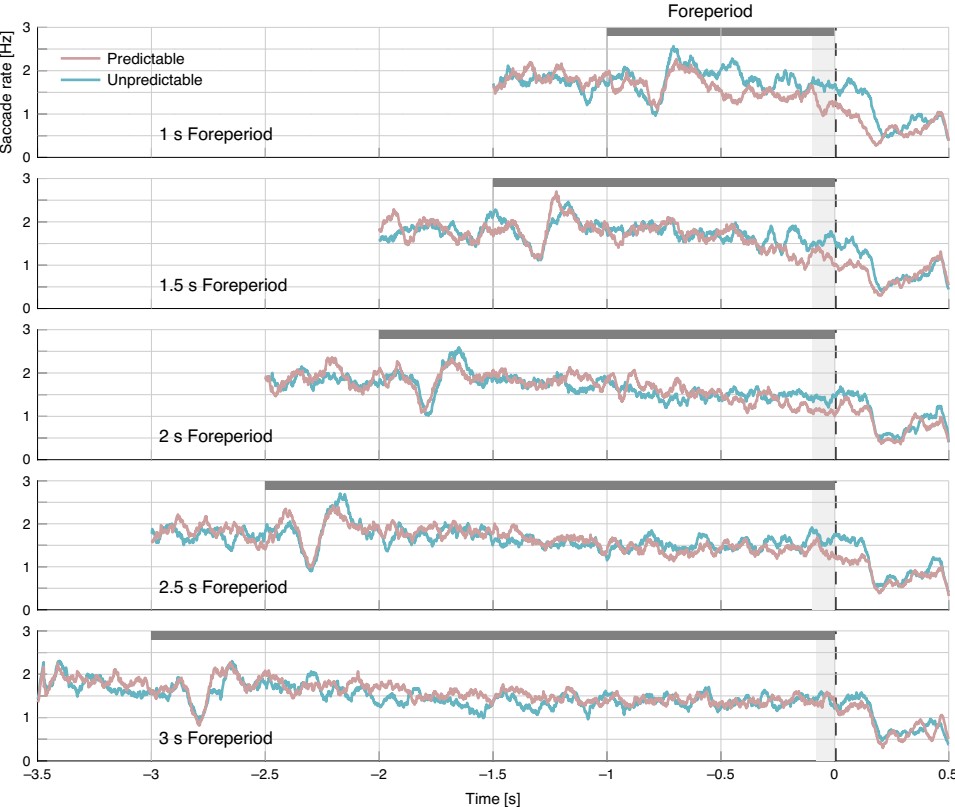

**Fig. 3 Saccade rates by predictability and foreperiod.** Grand average ($N = 20$) saccade rate traces in the predictable (pink) and unpredictable (turquoise) conditions in each foreperiod duration. The dark gray horizontal rectangle represents the foreperiod duration. The dashed line represents target onset. The light gray shading represents the analyzed interval. Source data are provided as a Source Data file.

having multiple types of predictable blocks. We focused on the shortest foreperiod of 1 s, as performance for short foreperiods is less affected by modulations that are due to the progress of time following the cue. It is well known that with variable foreperiods, reaction time is faster for longer foreperiods[12]. This effect is thought to be related to expectation modulations caused by changes across time in the probability of an event to occur given that it has not occurred yet (conditional probabilities).

**Behavioral performance: accuracy-rates and reaction times.**
Participants performed better when presented with 80% predictable than unpredictable targets (Fig. 7). Paired-samples $t$-test showed that participants had significantly higher accuracy rates in the 80% predictable (mean accuracy rate = 0.868, SD = 0.105) than in the unpredictable (mean accuracy rate = 0.826, SD = 0.104) condition ($t(19) = 3.031$, 95% CI = [0.013 0.071], $p = 0.007$, Cohen's $d = 0.677$). Similarly, participants responded significantly faster to 80% predictable (mean RT = 0.748, SD = 0.227) than to unpredictable (mean RT = 0.825 s, SD = 0.241) targets ($t(19) = -2.948$, 95% CI = [−0.133 −0.023], $p = 0.008$, Cohen's $d = -0.659$).

**Saccades. Pretarget saccade rate.** Participants were less likely to initiate a saccade prior to the target when it was anticipated with 80% probability to appear 1000 ms following the cue than when it was unpredictable. The smoothed saccade rate traces of both conditions are depicted in Fig. 8a. A paired-samples $t$-test confirmed that saccade rate at the analyzed interval (900–1000 ms relative to cue onset) was significantly lower in the 80% predictable condition (mean rate = 1.19, SD = 0.596) than in the unpredictable condition (mean rate 1.45, SD = 0.562; $t(19) = $

−2.904, 95% CI = [−0.448 −0.073], $p = 0.009$ Cohen's $d = -0.649$).

Saccade rate slope. Similarly, the slope of saccade rate following the cue presentation was steeper when a target onset was expected after 1 s. A paired-samples $t$-test confirmed that the saccade rate slope (the difference between the 900–1000 ms and the 400–500 ms post-cue rates) was significantly steeper in the predictable (mean slope = −0.845, SD = 1.01) than unpredictable (mean slope = −0.11, SD = 0.874) conditions ($t(19) = -2.648$, 95% CI = [−1.316 −0.154], $p = 0.016$, Cohen's $d = -0.59$).

**Blink rate. Pretarget blink rate.** Blinks were less likely to occur prior to target onset when it was anticipated at 80% chance than when it was unpredictable (Fig. 8b). Paired-samples $t$-test confirmed that, in the analyzed interval (500–1000 ms relative to cue onset), blink rate was significantly lower in the 80% predictable (mean rate = 0.067, SD = 0.073) than in the unpredictable (mean rate = 0.113, SD = 0.084) condition ($t(19) = -3.427$, 95% CI = [−0.075 −0.018], $p = 0.003$, Cohen's $d = -0.766$).

**Discussion**
Temporal predictability was manipulated by presenting either predictable or unpredictable targets in different blocks. In Experiment 1 the timing of predictable targets was 100% predictable and in Experiment 2 it was only 80% predictable. In both cases, even though cues and targets were auditory and there was no visual task other than maintaining fixation, saccades and blinks were reduced shortly prior to the onset and more so for predictable than unpredictable targets. Furthermore, in Experiment 1 we examined whether the evolution of this inhibition across time was also modulated by predictability, and found that

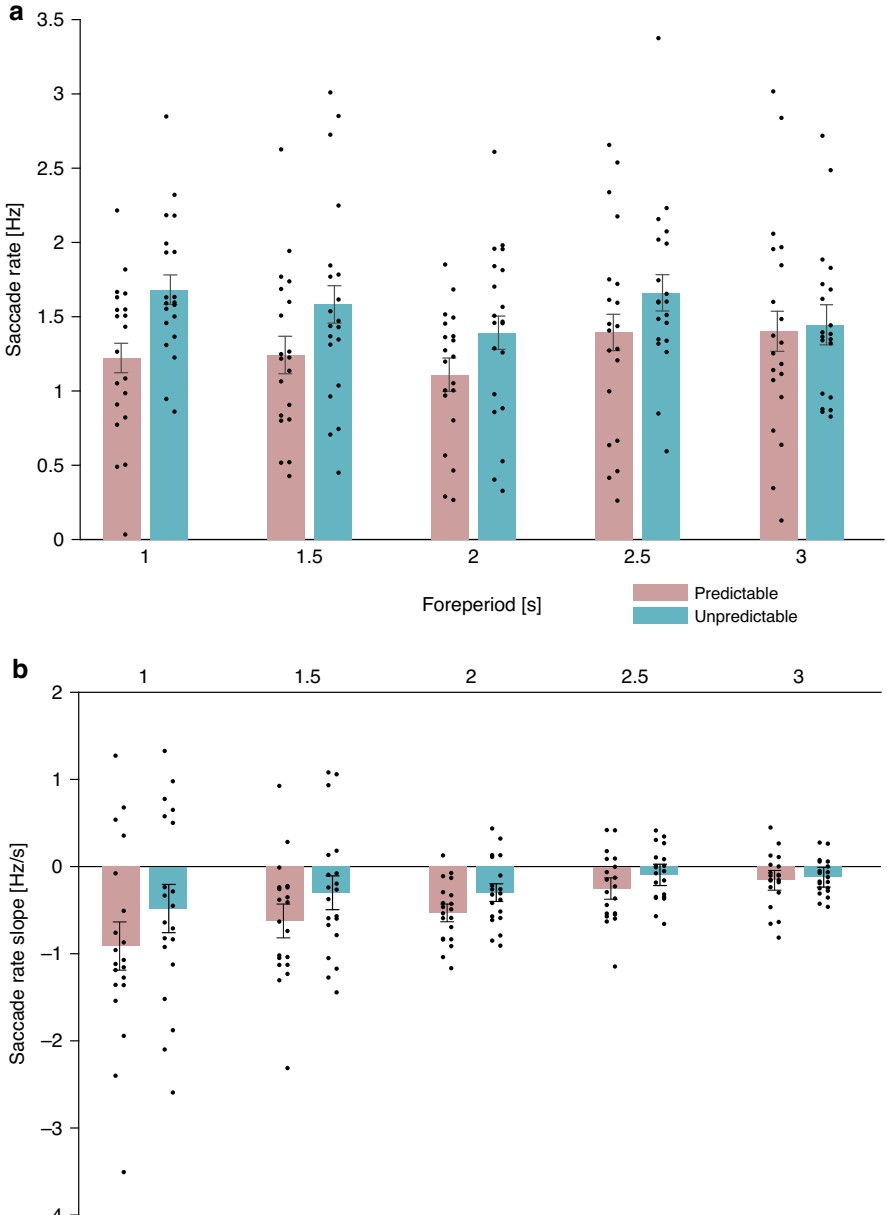

**Fig. 4 Pretarget saccade rates by predictability and foreperiod. a** Grand average pretarget saccade rate in the predictable (pink) and unpredictable (turquoise) conditions at −100 to 0 ms relative to target onset. $N = 20$ participants.; **b** Saccade rate slope assessed by calculating the normalized difference between saccade rate in the interval 400–500 ms following cue onset and saccade rate in the interval −100 to 0 ms relative to target onset at 0. This difference value was then divided by the time in seconds between the two intervals. Error bars denote ±1 standard error of the mean, corrected for within-subjects variability[60]. $N = 20$ participants. Source data are provided as a Source Data file.

the decrease in saccade rate prior to the onset of the target (slope) was steeper for predictable than unpredictable intervals in both experiments. In Experiment 2 we showed that the effect does not necessitate full certainty; it is induced also by probabilistic information, when there is only 80% probability for the predictable intervals. These results suggest that oculomotor activity was adjusted to reach a minimum at the onset of the anticipated auditory target. These findings, consistent with our results in the visual[3] and tactile[13] domains, reveal that the execution of oculomotor events is modulated by target's predictability, even when the target is auditory.

The present experiments revealed that oculomotor inhibition measurements reliably show a predictability effect, demonstrating its effectiveness in indexing temporal expectations and revealing a link between oculomotor behavior and auditory temporal

expectation. In contrast, accuracy and RT effects only emerged in Experiment 2, in which accuracy was higher and RT was faster in the predictable than the unpredictable conditions. Some studies have reported effects of temporal expectation on accuracy and RT[12,13,14,15] but others have failed to find such effects[3,16,17]. Some task demands and/or stimulus parameters may be responsible for these differences. Consistent with our previous study[3], the present findings support the hypothesis that oculomotor inhibition is a reliable index of predictability that is less affected by task demands and stimulus parameters.

The perceptual system is constantly exploring the environment. As humans, vision is our dominant source of input and eye movements are critical for exploration: We gather information on the surroundings by shifting our gaze from one location of interest to another. Visual exploration through eye movements is

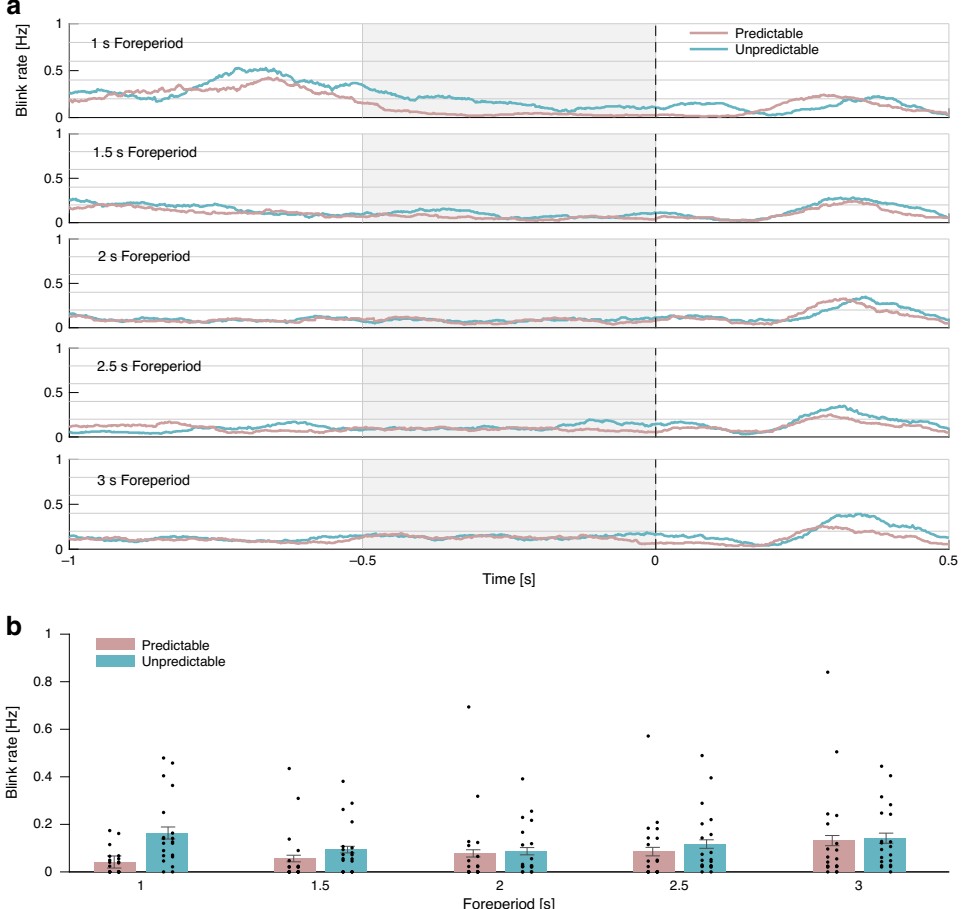

**Fig. 5 Blink rates by predictability and foreperiod. a** Grand average ($N = 20$) blink rate traces in the predictable (pink) and unpredictable (turquoise) conditions in each foreperiod duration, smoothed with a sliding window of 100 ms. The light gray rectangles mark the analysis window. The dashed line represents target onset. **b** Grand average pretarget blink rate in the predictable (pink bars) and unpredictable (turquoise bars) conditions at −500 to 0 ms relative to target onset. Error bars denote ±1 standard error of the mean, corrected for within-subjects variability[60]. $N = 20$ participants. Source data are provided as a Source Data file.

such a basic drive in humans that it occurs even when visual information is entirely irrelevant, such as when performing nonvisual tasks[18]. However, during the anticipation period, while the perceptual system prepares to process an upcoming target, it may be counterproductive to accumulate new inputs through active exploration. During this period, it may be advantageous to briefly pause exploration and focus resources on the anticipated stimulus. Our present findings of an inhibition of saccades prior to anticipated targets is consistent with this hypothesis, as they show that the freeze of visual exploration occurs even when anticipating an auditory stimulus.

The duration of the foreperiod—the interval between cue and target—is known to affect temporal expectations. When foreperiods are constant, longer foreperiods usually result in slower RTs, and when foreperiods are variable, longer foreperiods usually result in faster RTs[12,19,20]. In the visual modality[3], we found that oculomotor inhibition featured both expected trends across foreperiods. Pretarget saccade rate increased with foreperiod duration in the predictable condition and decreased in the unpredictable condition. In Experiment 1 we examined predictability effects across a range of foreperiods and found neither of these trends with saccades, and only the negative trend of the predictable condition with blinks (i.e., higher blink rate for longer foreperiods). These results may suggest that there are several subprocesses involved in temporal expectations: the basic anticipatory process that differentiates predictable and

unpredictable targets is effective for both visual and auditory targets, but other processes may be specific to the visual modality. It is also possible that the more subtle processes can be exposed only with higher statistical power.

In the unpredictable condition there is minimal certainty regarding the timing of the target, but it could be hypothesized that some statistical inference can, nevertheless, be used to estimate the onset of the target. In Experiment 1, in which predictability effects were examined over a range of foreperiods, the findings support this hypothesis by showing that in the unpredictable condition microsaccadic inhibition was maximal in the mean (and median) foreperiod of 2 s. These results suggest that, in the absence of accurate information, statistical inference regarding the mean foreperiod was used to estimate the onset of the target.

Why are saccades and blinks inhibited prior to the occurrence of a predictable target? One possibility is that this pretarget oculomotor inhibition serves a functional role in perceptual performance, i.e., that avoiding saccades and blinks while anticipating a predictable target enhances subsequent target perception. This hypothesis is plausible when considering visual targets and tasks. Saccades and blinks are known to cause a temporary loss of visual input due to physical occlusion, image blur or masking[21,22] and also be accompanied by active suppression in sensory cortices (blink suppression[23] and saccadic suppression[24]). Consistently, in our previous study on temporal

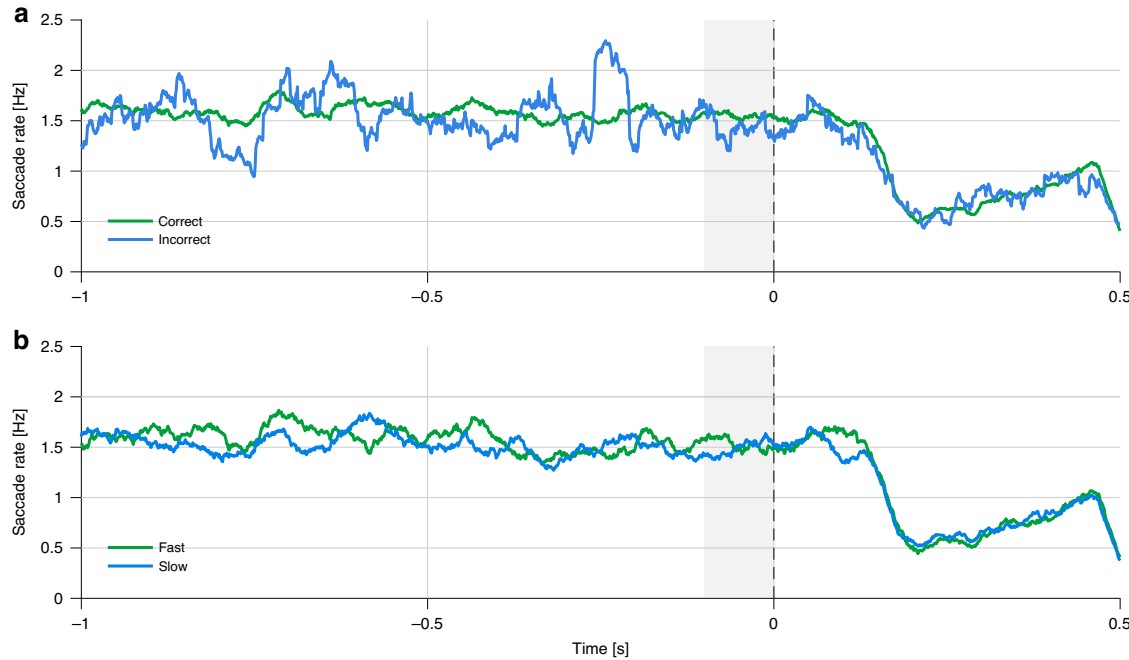

**Fig. 6 Saccade rates according to trial performance. a** Grand average saccade rates in correct trials (green) and incorrect trials (blue) of the unpredictable condition. The dashed line represents target onset. The gray rectangle marks the pre-stimulus analysis window. **b** Grand average saccade rates in fast trials (green) and slow trials (blue) of the unpredictable condition, divided according to the median. Source data are provided as a Source Data file.

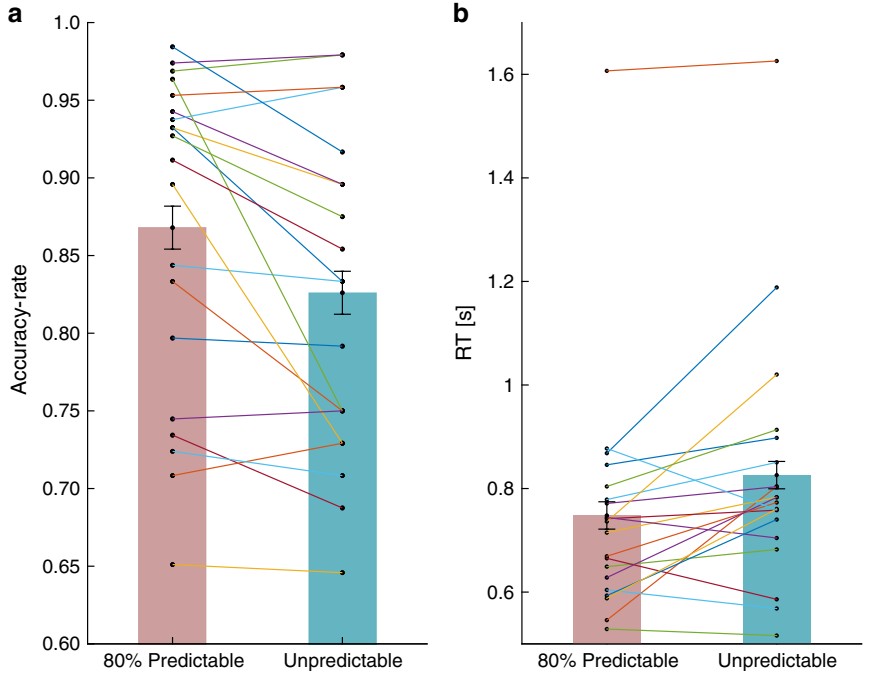

**Fig. 7 Experiment 2: Accuracy-rates and reaction times (RTs) in the 1 second foreperiod. a** Accuracy-rates in predictable (pink bars) and unpredictable (turquoise bars) conditions. $N = 20$ participants. **b** Reaction times in predictable (pink bars) and unpredictable (turquoise bars) conditions. Error bars denote ±1 standard error of the mean, corrected for within-subjects variability[60]. The colored connecting lines represents individual participants. $N = 20$ participants. Source data are provided as a Source Data file.

expectations in the visual domain, we found decreased accuracy rates and increased RTs when saccades were performed during target presentation[3]. Notably despite the fact that the observed oculomotor inhibition in that study lasted for a few hundred milliseconds prior to target onset, we did not find any perceptual advantage for inhibiting oculomotor events that did not overlap

with target presentation. This longer inhibition period could nevertheless serve a functional role as it may reduce the likelihood that an oculomotor event would occur around the time of target presentation.

Remarkably, we find oculomotor inhibition prior to auditory targets, in the absence of any visual event. It is unlikely that eye

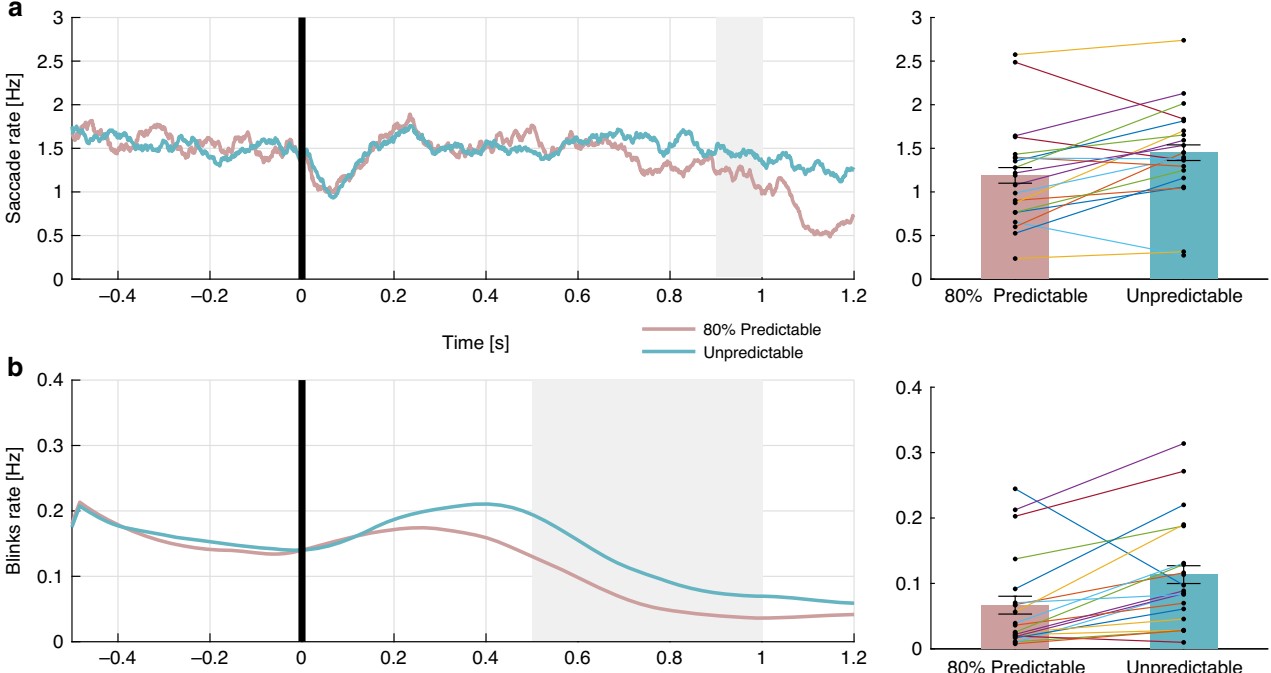

**Fig. 8 Experiment 2: Saccade rates and blink rates. a** Grand average saccade rate traces in the 80% predictable (pink) and the unpredictable (turquoise) conditions. The gray rectangle marks the 900–1000 ms post-cue analysis window. The black line represents cue offset. The bar graph to the right depicts the calculated saccade rate average within the analysis window. $N = 20$ participants. **b** Grand average of the blink rates traces in the 80% predictable (pink) and the unpredictable (turquoise) conditions. The gray rectangle marks the 500–1000 ms post-cue analysis window. The black line represents cue offset. The bar graph to the right represents the calculated blink rate average within the analysis window. Error bars denote ±1 standard error of the mean, corrected for within-subjects variability[60]. The colored connecting lines represent individual participants. $N = 20$ participants. Source data are provided as a Source Data file.

movements would cause any loss of input, as most sources of oculomotor interference (occlusion, blur and masking) do not affect perception in nonvisual modalities. It is possible, in principle, that active cortical suppression during saccade and blinks would affect not only the visual cortex but also other sensory cortices, either by crossmodal interactions or by global mechanisms. However, there is currently no evidence supporting the existence of such an effect, and indeed, the detection of auditory targets is not affected by concurrent saccades[25]. In the present study, we found no behavioral cost for executing a saccade, even during target presentation, as would be predicted if active cortical suppression was involved (but see different findings in the tactile domain[13]). These findings suggest that, regardless of the mechanism that drives this effect, oculomotor inhibition prior to predictable stimulation does not occur solely for functional advantages.

The present study reveals a correlate of temporal expectations, by showing that oculomotor inhibition is present prior to auditory targets. This inhibition emerged even in a task in which performance was completely unaffected by the execution of oculomotor events. These findings are consistent with studies showing other multimodal aspects of temporal expectations: (1) an event related potentials (ERP) study in which temporal attention affected the early post-target ERP components of both tactile and visual responses, regardless of the modality of the specific task[26]; (2) Participants performed better in visual tasks when the visual target appearance was synchronized with the beat of an irrelevant auditory rhythm[27,28]. Beyond these studies showing post-target consequences of temporal expectations, the present study reveals that perceptual expectation is tightly coupled to oculomotor action.

In this study we used a common design for studying expectations, in which predictability is manipulated across blocks[3,20,29–32].

With this type of design, preparation effects may reflect both an intentionally driven preparatory process guided by expectancies and an unintentional process that is based on a conditioned response elicited by the cue[33,34]. According to the conditioning view, predictability effects are due to 'trace conditioning'—a conditioned response that is time locked to a conditioning stimulus (the cue) and peaks around the time of the conditioned stimulus (the target). In the predictable condition, the constant interval between the cue and the target is repeatedly reinforced, while other intervals are suppressed. In contrast, in the unpredictable blocks with varied intervals there is no continuous reinforcement. It is unknown whether conditioning is involved in the oculomotor inhibition effect, yet it has already been determined that conditioning is not the sole explanation for the temporal orientation effects in RTs[35]. Given that saccades may be performed either voluntarily or involuntarily, the link revealed in this study between saccadic inhibition and temporal expectation is consistent with a combination of intentional and unintentional processes in mediating temporal expectations.

The link between temporal expectation and oculomotor inhibition is likely mediated by an interaction of cortical and subcortical structures, consistent with the possibility of both intentional and unintentional processes. For example, the dorsolateral prefrontal cortex (DLPFC) is involved in various tasks of temporal expectation and timing of intervals[36–40], and has extensive direct and indirect connections to the main cortical and midbrain oculomotor areas[41]. The DLPFC also contains neurons that directly project to the superior colliculus (SC), a midbrain region that controls saccadic eye movements[42], which is connected to oculomotor cortical areas, such as the frontal eye field (FEF), the supplementary eye field (SEF) and the parietal eye field (PEF)[41,43,44]. The DLPFC is specifically involved in saccadic

inhibition, which is mediated by the direct connection to the SC through the prefrontal-collicular tract[44], and by the indirect connection to the SC via the basal ganglia[45,46]. These areas may also be involved in the oculomotor inhibition mechanism of temporal expectations. It is unlikely that only subcortical structures mediate the oculomotor inhibition effect, as the responsible structures should enable the perception and retention of the duration of intervals. Whereas the autonomic system, previously associated with expectations[47], is unlikely to have this timing abilities, the cerebellum may be a relevant structure as it has been implicated in the formation of cue-based expectations[48] and in conditioning[49].

Importantly, regardless of whether oculomotor inhibition is driven by a bottom-up, a top-down, or both mechanisms, and regardless of whether it involves subcortical, cortical, or both regions, the present findings reveal that it is tightly linked to temporal expectations, and that this link goes beyond a mere functional role of preventing negative effects of saccadic movements and the corresponding blur on visual perception.

Brain regions that are involved in the oculomotor inhibition effect, may be either part of a crossmodal or a supramodal system. The crossmodal hypothesis suggests that oculomotor inhibition prior to auditory targets is the result of crossmodal interactions between the two sensory modalities. According to this view, the visual system prepares for an upcoming predictable event, even when this event is not visual. This visual preparation is reflected by a reduction in the number of eye movements prior to a predictable auditory target. This view is supported by behavioral and neurophysiological findings suggesting that there are wide-spread crossmodal links between the visual and the auditory systems, some of which involve the oculomotor system[7,50,51]. In contrast, the supramodal hypothesis suggests that the oculomotor inhibition reflects a supramodal control mechanism of temporal expectation: a mechanism that is neither visual nor auditory but is involved in the formation of temporal expectations in both modalities. This view is supported by behavioral evidence showing that, in certain contexts, oculomotor behavior is modulated by nonsensory mechanisms that are not directly related to the visual system[10,18].

To conclude, oculomotor inhibition reliably captures the existence of temporal prediction, regardless of the presence or absence of other behavioral predictability effects. The pretarget oculomotor inhibition marker of temporal expectations reflects the formation of expectations rather than their outcome; therefore, it is influenced solely by early pretarget processes and less sensitive to specific stimulus parameters, instructions and criterion. Together with the corresponding findings in the visual domain[2,3] and the tactile domain[13], the present findings indicate that pretarget oculomotor inhibition is a marker of temporal expectation across vision, touch, and audition. These findings reveal how our very basic human drive to explore can be momentarily paused in anticipation for an upcoming event of interest, even when this event will be processed via a different modality.

## Methods

**Experiment 1**. Subjects: Twenty-one students of Tel Aviv University participated in the experiment in exchange for course credit or monetary compensation. One participant was discarded from all analysis due to failure to comply with the task. Consequently, eye tracking and behavioral analysis were based on a total of 20 participants (14 females; mean age 22.9 ± 2.7). The sample size of $N = 20$ was determined following a power analysis simulation described below.

All participants reported normal (uncorrected) vision and audition and no history of neurological disorders. All were naïve to the purpose of this study. The ethical committees of Tel Aviv University and the School of Psychological Sciences approved the study. All participants signed an informed consent.

Power analysis stimulation: To determine the required number of participants that will lead to power of 80% using a two-tailed criterion of .05, we conducted a

simulation based on data of our previous study[3] ($N = 20$). Datasets were iteratively sampled (without replacement) to create random samples with sizes ≥5. For each sample size, resampling was based on 10,000 iterations. We conducted a $2 \times 5$ repeated-measures ANOVA on the data-set produced by each iteration, using the same factors as in the current experiment, and extracted the $p$ value for Predictability (predictable/ unpredictable). For each sample size, we then calculated the null rejection proportion (i.e., power) out of all iterations. A sample size of 12 participants led to this result ($1 - \beta = 0.86$), confirming that a cohort of 20 participants would be large enough to achieve reliable results with these effect sizes.

Stimuli: The cue was a pure tone of 5 KHz, played for 33 ms. The target tone was a descending or ascending chirp sound lasting 33 ms, constructed from a linear swept-frequency pure tone, starting or ending at 800 Hz. A short pretest was conducted to set the difference between the two pitches of the chirp sound for each participant. Using a 1-up/2-down staircase procedure[52], we aimed to obtain 70% accuracy rate. Following this procedure, the average other, higher pitch was 940.7 Hz (SD 89.91 Hz). The two pitches of the chirp sound remained constant throughout the experiment. All sounds were played binaurally over headphones (Audio-Technica ATH-M50x).

Procedure: Participants sat, head resting on a headrest in a dimly lit sound-attenuated chamber, at a distance of 97 cm from a display monitor (ASUS VG248QE, 120 Hz refresh rate) covering 30° of the horizontal visual field. In each trial, a black fixation cross (0.4°) was centrally presented on a mid-gray background. Participants were instructed to maintain fixation throughout the trial duration. After an online gaze contingent procedure confirmed fixation (<0.5° off center) and following an additional random interval (0.4–0.9 s), the temporal cue was played for 33 ms, marking the onset of the foreperiod (1/1.5/2/2.5/3 s). After the foreperiod, the target tone was played for 33 ms and participants were asked to perform a 2-alternative forced choice (2AFC) discrimination task: report whether the chirp was ascending or descending by pressing one of two buttons. We instructed participants to be as accurate as possible and to respond within the 4 seconds response window. Following the response, or after 4 s without one, the fixation cross changed color to gray for 200 ms to signal the end of the trial. Figure 1 depicts the trial sequence.

The foreperiod was either constant throughout the block (predictable condition) or changed randomly in different trials within the same block (unpredictable condition). Thus, the cue acted as a 100% valid temporal cue in the predictable condition but was uninformative regarding target timing in the unpredictable condition. Importantly, the stimuli were identical in the two conditions, and differed only in the validity of the temporal cue in predicting the time of the target. Participants were not informed as to any predictability; therefore, all temporal expectations were learned incidentally. The experimental session was divided into 10 blocks of 100 trials per block, lasting ~6.45 min each, half of which corresponded to the predictable condition and half to the unpredictable condition. The order of the blocks was counterbalanced across participants. There was an 8 minutes break after 5 blocks, and shorter breaks between blocks, when necessary.

Behavioral data analysis: Accuracy-rates and reaction times (RT) were calculated separately for each participant, condition and foreperiod. Only correct trials were included in the RT analysis. Outlier RTs deviating by more than 2.5 standard deviations (SD) from the mean RT were excluded from analysis.

Eye tracking acquisition and analysis: Binocular gaze position was monitored using a remote infrared video-oculographic system (Eyelink 1000 Plus; SR Research, Canada), with a spatial resolution ≤0.01° and average accuracy of 0.25°–0.5° when using a headrest (as reported by the manufacturer). Raw gaze positions were converted into degrees of visual angle using the 9-point-grid calibration, performed at the start of each experimental block and sampled at 1000 Hz.

Blinks were detected using the Eyelink's algorithm. Saccades were detected using a modification of a published algorithm[53], which was applied on filtered gaze position data (low-pass IIR Butterworth filter; cutoff 60 Hz; as in Amit, Abeles, Bar-Gad, & Yuval-Greenberg, 2017[54]). An elliptic threshold criterion for microsaccades detection was determined in 2D velocity space based on the horizontal and the vertical velocities of the eye-movement. Specifically, we set the threshold to be six times the SD of the eye-movement velocity, using a median-based estimate of the SD[55]. The SD estimate was set based on the recordings of each trial. A saccade onset was defined when six or more consecutive velocity samples were outside the ellipse, in both eyes.

Saccades offsets are sometimes accompanied by an overshoot, which may be erroneously detected as a new saccade. Therefore, per standard procedure[56–58], we imposed a minimum criterion of 50 ms for the interval between two consecutive saccades and kept only the first saccade in cases where two saccades were detected within such interval. Saccades of all sizes were included in the analysis, but due to the instruction to keep sustained fixation, most (mean 87.6%, SD 10.3%) saccades were small (in the range of microsaccades, <1°)[59].

The time series of saccade rate and blink rate were constructed for each participant by counting the number of saccade/blink events in each time-point across trials, separately for each condition and foreperiod, and dividing these values by the number of trials. The saccade time series was then smoothed using a sliding window of 50 ms, and multiplied by the sampling rate, converting the measure to Hz. Following our previous studies[2,3], mean saccade rate in the time window of −100–0 ms relative to target onset was taken as the dependent variable for

statistical analysis of pretarget saccade rate (PSR). This time interval was chosen to assess saccade rate shortly prior to target onset. Since blink events are sparse and last longer, the blink rate time series was smoothed using a sliding window of 100 ms and averaged across a longer window of −500 to 0 ms relative to target onset and multiplied by the sampling rate to convert to Hz. Saccade rate slope was calculated as the difference between saccade rate at the pretarget window (−100 to 0 ms relative to target onset) and the post-cue window (400–500 ms post-cue onset, after the saccade rate returns to baseline following the cue presentation, a microsaccade-rate signature[5]) divided by the time difference in seconds between the two windows (which was different for each foreperiod duration; as in Amit et al.[3]).

Statistical analysis: In Experiment 1, most statistical analyses were based on repeated-measures ANOVAs with factors Predictability (predictable/unpredictable) and Foreperiod (1/1.5/2/2.5/3 s). Significant interactions were followed up by trend analysis testing for linear and quadratic trends. The assumption of sphericity was tested, when applicable, using Mauchly's test. When Mauchly's test was significant ($p < 0.05$) the Greenhouse–Geisser corrected $p$-values are reported, along with the original degrees of freedom and the epsilon value. All statistical tests performed were two-tailed.

**Experiment 2**. Subjects: Twenty-two students of Tel Aviv University participated in Experiment 2. Two participants were excluded from the experiment due to ceiling performance on the task (more than 2 blocks with 100% accuracy). Consequently, eye tracking and behavioral analysis were based on a total of 20 participants (13 females; mean age 24.9 ± 4.37). All participants reported normal (uncorrected) vision and audition and no history of neurological disorders. All were naïve to the purpose of this study. The ethical committees of Tel Aviv University and the School of Psychological Sciences approved the study. All participants signed an informed consent.

Stimuli: As described in Experiment 1.

Procedure: The procedure was the same as in Experiment 1, except that in the predictable blocks the majority of trials (80%) included a foreperiod of 1 s and only a minority (20%) included a foreperiod of 2.2 s. The unpredictable blocks were identical to those of Experiment 1 (foreperiods 1–3 s in 0.5 s steps with equal probabilities). The experimental session was divided into six blocks (3 predictable blocks and 3 unpredictable) of 80 trials

Eye tracking acquisition and analysis: Analysis in this experiment focused on the 1 s intervals following the cue, which was the only predictable foreperiod used in this experiment. Consequently, in the behavioral analysis only trials with foreperiod 1 s were included. In the eye tracking analysis, we collapsed the data across all the unpredictable foreperiods and analyzed the 1 s interval following the cue. The dependent variables were, therefore, the mean saccade rate at 900–1000 ms and blink rate at 500–1000 ms following cue onsets regardless of actual foreperiod duration.

As in Experiment 1, most saccades were smaller than 1° (mean 89.4%, SD 12.9%).

**Reporting summary**. Further information on research design is available in the Nature Research Reporting Summary linked to this article.

## Data availability

The data supporting the findings of this study, all custom scripts and the source code for Figs. 2–8 have been made available at the open science framework with the identifier: https://doi.org/10.17605/OSF.IO/X7RSD. A reporting summary for this Article is available as a Supplementary Information file.

## Code availability

The custom code used in the analysis is available at the open science framework with the identifier: https://doi.org/10.17605/OSF.IO/X7RSD

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

## Acknowledgements

We thank Noam Shimoni for his assistance in running the experiment, and Stephanie Badde for useful discussions. This study was funded by the United States-Israel Binational Science Foundation grant 2015201 to S.Y.-G and M.C.

## Author contributions

D.A., R.A., M.C., and S.Y.-G. designed this research. D.A. and R.A. performed the experiments. D.A., R.A., and N.T. analyzed the data. D.A., M.C., and S.Y.-G. wrote the manuscript.

## Competing interests

The authors declare no competing interests.
