## [Peer Review File · Nature Communications]

Reviewers' Comments:

Reviewer #1:

Remarks to the Author:

Consistent with previous work, the authors demonstrate that saccade execution is modulated by the predictability of a target. In a novel expansion of prior studies, this manuscript presents a case in which these relationships exist in response to auditory (rather than visual) stimuli. Thus, saccades are inhibited before auditory targets are expected. This is high-quality research that provides further evidence that the oculomotor system is recruited in circumstances where visual information is irrelevant. The work has been conducted to a high methodological standard, the findings appear sound, and the conclusions are appropriate. The article is well-written, clear, professional, and detailed.

Minor comments:

1. One small point about the language used: Describing the inhibition as a "supramodal inhibitory mechanism" that "goes beyond vision" seems problematic, because the evidence points to a mechanism inhibiting the oculomotor system and not the auditory system. The inhibition itself is not supramodal here, though it is in response to a supramodal anticipatory process. In many places (for example, lines 406 and after), the authors describe the findings as oculomotor inhibition and a supramodal temporal expectation, which fits my understanding of their discoveries.
2. Were participants instructed to maintain gaze on the fixation cross for the full duration of the trial period?
3. Lines 72-73 (and perhaps also after Line 169): it would be appropriate to include a reference to prior work showing modulation of microsaccade dynamics during other non-visual tasks, such as mental arithmetic, see Siegenthaler et al, Eur J. Neurosci. 2014.
4. Line 169: "in the range of microsaccades, <1 degree." Here it would be appropriate to include a citation to Martinez-Conde et al, TINS 2009, who (to my knowledge) first formalized the argument that one should consider microsaccade sizes to extend up to 1 degree (see Box 1: 'How large is a microsaccade?').
5. There are some strange characters on line 340.

Reviewer #2:

Remarks to the Author:

I think that this is an interesting study addressing a potentially important issue about temporal expectation. The paper very well written, the experiment is conducted with care and the data analysis is appropriate. The theoretical impact of the paper is less clear-cut as there is no obvious theoretical reason to expect oculomotor inhibition prior to auditory targets. Although there is much to like about this paper, there is one fundamental issue that needs to be resolved.

As the main message the paper claims that "These findings document a novel link between oculomotor inhibition and temporal processing and thus reveal a global supramodal mechanism of temporal expectations" (abstract). It is argued that "temporal expectations are formed by the cognitive system" (p.2). The question is whether this claim really holds given the experimental paradigm. My main concern is the use of the notion of "temporal expectation". The paradigm generates a particular rhythm of tones that are precisely presented at fixed intervals of one another. What happens is that the human body reacts to this rhythmic sequence generating all kind of physiological reactions that may coincide with these events. This may be heart rate variability, blood volume pulse, GSR and maybe, as in the current case, eye movements. This is interesting but do we believe that these automatic physiological reactions of the body represents anything about "expectation". I do not think so and I do not think that such physiological reactions are that interesting. Now, there are way to really show that it is not some kind of low level autonomic reaction

of the nervous system but instead represents true "expectation". The experiment that needs to be done is the following: Instead of having one constant foreperiod during a whole block there should be two foreperiods of different length; each foreperiod is announced by a 100% valid cue (for example a high tone for foreperiod 1 sec and a low tone for foreperiod 2.5 sec). If it is really temporal expectation then we expect to see basically the same results as what was shown here. The cue generates an expectation for a particular interval and we should see this back in the eye movement data. If it is just a low level physiological reaction we expect that the results will not hold. This experiment seems to be crucial and without this experiment the effect reported here may be a trivial physiological reaction of the human body to a rhythmic tone.

Reviewer #3:

Remarks to the Author:

Frequency of saccades and blinks reduce prior to expected onset of visual targets, a process assumed to be used by the brain to improve the detection/processing of newly presented visual information. Abeles and colleagues use a psychophysical paradigm to show that this oculomotor inhibition does also occur prior to the onset of auditory stimuli. This suggests that rather than a modality-specific system, the oculomotor inhibition relies on a multi-modal module.

Major comments:

1) It is not clear to me how this new finding is changing our perspective regarding the purpose of oculomotor inhibition. As pointed out by the authors the exact mechanism by which the "expectation" alters performance is not clear but I am not aware of any study showing that expectations stay within their specific modality and are not transferred across modalities. In other words, at least based on the literature cited in the paper and my knowledge, the results were not against any dogma and were expected.

2) The authors mention that "Finding no oculomotor inhibition prior to predictable auditory targets would support the hypothesis that the functional role of the oculomotor inhibition is merely to facilitate vision. Alternatively, finding an oculomotor inhibition effect for auditory targets would imply that its functional role goes beyond vision."

If "it" in the last sentence refers to "oculomotor inhibition", then the rationale is not true. The alternative result does not essentially mean that functional role of oculomotor inhibition goes beyond vision. It could simply mean that "expectation" causes oculomotor inhibition to improve visual performance and at the same time alters processing in other domains to improve performance in other modalities.

Therefore, my main concern is that the results do not provide any insight regarding the functional significance of oculomotor inhibition, except showing that its controlling "expectation" module controlling is a supramodal one, which was expected (or not counter-argument was provided).

Minor:

1) Why would the authors choose 400-500 ms after cue onset as the post cue window? They can slide a window (with different widths) through time to see which period shows most significant effect.

2) There is no reference to figure 6 in the text.

3) Page 7, line 177: "asses" should read assess.

4) Page 10, line 240: "figure 3C" is wrong. It should be fixed.

5) Y-axis in figures 5A and B should be blink rate.

6) Caption in Figure 5B: "Grand average pre-target saccade rate" is wrong. It should be "Grand average pre-target blink rate".

7) In order to have a better illustration for all the figures, I think it is a good idea to use red vs blue rather than purple vs blue.

Reviewer #1:

Consistent with previous work, the authors demonstrate that saccade execution is modulated by the predictability of a target. In a novel expansion of prior studies, this manuscript presents a case in which these relationships exist in response to auditory (rather than visual) stimuli. Thus, saccades are inhibited before auditory targets are expected. This is high-quality research that provides further evidence that the oculomotor system is recruited in circumstances where visual information is irrelevant. The work has been conducted to a high methodological standard, the findings appear sound, and the conclusions are appropriate. The article is well-written, clear, professional, and detailed.

— Thanks for your positive evaluation and constructive comments; addressing them has strengthened our manuscript.

Minor comments:

1. One small point about the language used: Describing the inhibition as a “supramodal inhibitory mechanism” that “goes beyond vision” seems problematic, because the evidence points to a mechanism inhibiting the oculomotor system and not the auditory system. The inhibition itself is not supramodal here, though it is in response to a supramodal anticipatory process. In many places (for example, lines 406 and after), the authors describe the findings as oculomotor inhibition and a supramodal temporal expectation, which fits my understanding of their discoveries.

— We have revised this throughout the manuscript accordingly [Lines 400, 409-412,32-36]

2. Were participants instructed to maintain gaze on the fixation cross for the full duration of the trial period?

— We have specified the instructions [lines 108-109]

3. Lines 72-73 (and perhaps also after Line 169): it would be appropriate to include a reference to prior work showing modulation of microsaccade dynamics during other non-visual tasks, such as mental arithmetic, see Siegenthaler et al, Eur J. Neurosci. 2014.

— We have included such references [lines 68-69]

4. Line 169: “in the range of microsaccades, <1 degree.” Here it would be appropriate to include a citation to Martinez-Conde et al, TINS 2009, who (to my knowledge) first formalized the argument that one should consider microsaccade sizes to extend up to 1 degree (see Box 1: ‘How large is a microsaccade?’).

— We have included this citation [lines 166]

5. There are some strange characters on line 340.

— We have deleted those characters [Line 337]

Reviewer #2:

I think that this is an interesting study addressing a potentially important issue about temporal expectation. The paper very well written, the experiment is conducted with care and the data analysis is appropriate. The theoretical impact of the paper is less clear-cut as there is no obvious theoretical reason to expect oculomotor inhibition prior to auditory targets.

— Thanks for your positive evaluation. Clarifying the issues you pointed out below has strengthened our manuscript. Moreover, we have clarified that our finding has theoretical relevance, notwithstanding the fact that there were no obvious reasons either to expect it or to not expect it.

Although there is much to like about this paper, there is one fundamental issue that needs to be resolved. As the main message the paper claims that “These findings document a novel link between oculomotor inhibition and temporal processing and thus reveal a global supramodal mechanism of temporal expectations” (abstract). It is argued that “temporal expectations are formed by the cognitive system” (p.2). The question is whether this claim really holds given the experimental paradigm. My main concern is the use of the notion of “temporal expectation”. The paradigm generates a particular rhythm of tones that are precisely presented at fixed intervals of one another. What happens is that the human body reacts to this rhythmic sequence generating all kind of physiological reactions that may coincide with these events. This may be heart rate variability, blood volume pulse, GSR and maybe, as in the current case, eye movements. This is interesting but do we believe that these automatic physiological reactions of the body represents anything about “expectation”. I do not think so and I do not think that such physiological reactions are that interesting.

Now, there are ways to really show that it is not some kind of low level autonomic reaction of the nervous system but instead represents true “expectation”. The experiment that needs to be done is the following: Instead of having one constant foreperiod during a whole block there should two foreperiods of different length; each foreperiod is announced by a 100% valid cue (for example a high tone for foreperiod 1 sec and a low tone for foreperiod 2.5 sec). If it is really temporal expectation then we expect to see basically the same results as what was shown here. The cue generates an expectation for a particular interval and we should see this back in the eye movement data. If it is just a low level physiological reaction we expect that the results will not hold. This experiment seems to be crucial and without this experiment the effect reported here may be a trivial physiological reaction of the human body to a rhythmic tone.

- Please note that the procedure we used in this study does not produce a rhythm of tones. Our design is a *non-rhythmic* design based on the presentation of a cue followed by a predictable or non-predictable target with a variable ISI between trials. We further clarify this issue in the Introduction [lines 71-72].
- We have clarified the logic of this study and stressed the novelty of the findings to highlight its theoretical relevance [lines 70-72; 409-412].

Reviewer #3:

Frequency of saccades and blinks reduce prior to expected onset of visual targets, a process assumed to be used by the brain to improve the detection/processing of newly presented visual information. Abeles and colleagues use a psychophysical paradigm to show that this oculomotor inhibition does also occur prior to the onset of auditory stimuli. This suggests that rather than a modality-specific system, the oculomotor inhibition relies on a multi-modal module.

- Thanks for your comments. Clarifying the following points in response to your comments has strengthened our manuscript.

Major comments:

1) It is not clear to me how this new finding is changing our perspective regarding the purpose of oculomotor inhibition. As pointed out by the authors the exact mechanism by which the “expectation” alters performance is not clear but I am not aware of any study showing that expectations stay within their specific modality and are not transferred across modalities. In other words, at least based on the literature cited in the paper and my knowledge, the results were not against any dogma and were expected.

- This is the first study to investigate this issue. Thus, the results cannot be in agreement or against any dogma; however, they were not expected (as we indicated and Reviewer 2 points out). We have clarified the logic of this study and stressed the novelty of the findings to highlight its theoretical relevance [lines 54-55; 61-62; 66-72; 409-412].

2) The authors mention that “Finding no oculomotor inhibition prior to predictable auditory targets would support the hypothesis that the functional role of the oculomotor inhibition is merely to facilitate vision. Alternatively, finding an oculomotor inhibition effect for auditory targets would imply that its functional role goes beyond vision.”

If “it” in the last sentence refers to “oculomotor inhibition”, then the rationale is not true. The alternative result does not essentially mean that functional role of oculomotor inhibition goes beyond vision. It could simply mean that “expectation” causes oculomotor inhibition to improve visual performance and at the same time alters processing in other domains to improve performance in other modalities. Therefore, my main concern is that the results do not provide any insight regarding the functional significance of oculomotor inhibition, except showing that the its controlling “expectation” module controlling is a supramodal one, which was expected (or not counter-argument was provided).

— We have clarified the logic of this study and stressed the novelty of the findings [lines 409-412].

Minor:

1) Why would the authors choose 400-500 ms after cue onset as the post cue window? They can slide a window (with different widths) through time to see which period shows most significant effect.

— We have clarified this choice [lines 179-180]. Please note that sliding a window would not necessarily be a better alternative; to estimate the slope it is necessary to calculate two separated time intervals.

2) There is no reference to figure 6 in the text.

— We have added the reference in the text [lines 297-299].

3) Page 7, line 177: “asses” should read assess.

— We have fixed this typo [Line 174]

4) Page 10, line 240: “figure 3C” is wrong. It should be fixed.

— We have fixed this information [Line 239]

5) Y-axis in figures 5A and B should be blink rate.

— We have fixed this information [Figures 5A and B]

6) Caption in Figure 5B: “Grand average pre-target saccade rate” is wrong. It should be “Grand average pre-target blink rate”.

— We have fixed this information [Figure 5B caption – Line 278]

7) In order to have a better illustration for all the figures, I think it is a good idea to use red vs blue rather than purple vs blue.

— We have updated the colors in all the figures.

Reviewers' Comments:

Reviewer #1:

Remarks to the Author:

The authors have addressed all my previous comments. Congratulations on an excellent study!

Reviewer #2:

Remarks to the Author:

The authors choose to not address my concern claiming that I did not understand the experiment. I did understand the experiment (The figure with the method was clear enough) and there is a rhythmic sequence of cue and target (irrespective of the ISI, the target is always followed by the cue at a particular fixed time interval). I believe it is an autonomic reaction of the nervous system that has nothing to do with "expectation". The experiment I suggested would have solved this but the authors choose not to carry out this experiment. That is too bad because only that experiment would have told us whether "temporal expectations by the cognitive system" are really involved.

Reviewer #3:

Remarks to the Author:

Unfortunately, I am not able to find the answer to my concern in the rebuttal or the modified article. My concern is that the results were expected. An anticipated auditory stimulus prepares the attentional system through which it can pause other modalities and actions in order to better process the upcoming stimulus. In response to the same objection from Reviewer 2, the authors argue that since their design is not rhythmic it is not drawing an autonomic reaction of the nervous system. I would like to draw their attention to the findings of Zhao, Al-Aidroos, Turk-Browne that "Attention is spontaneously biased toward regularities." The regularity within the predictable condition is all what it takes to prepare the subjects to draw attention and I couldn't expect any result except what is shown.

I can completely understand that maybe there are crucial aspects that I am missing and other reviewers see that. I just couldn't imagine seeing any other result: an anticipated stimulus is expected to draw attention and takes the sensory resources toward its processing and put a dent in the processing of other signals and execution of motor commands. I would leave it for other reviewers and editors to decide if the paper is beyond this result or if this result was not known already.

Reviewer #1

The authors have addressed all my previous comments. Congratulations on an excellent study!
– Thank you for your support.

Reviewer #2

The authors choose to not address my concern claiming that I did not understand the experiment. I did understand the experiment (The figure with the method was clear enough) and there is a rhythmic sequence of cue and target (irrespective of the ISI, the target is always followed by the cue at a particular fixed time interval). I believe it is an autonomic reaction of the nervous system that has nothing to do with “expectation”. The experiment I suggested would have solved this but the authors choose not to carry out this experiment. That is too bad because only that experiment would have told us whether “temporal expectations by the cognitive system” are really involved.

– Thank you for your comments.

We respectfully disagree with the claim that the procedure is rhythmic. A rhythm should contain at least three consecutive repetitions of an interval pattern. The fact that in our procedure there was a fixed interval between the cue and the target makes this a *predictable or regular* condition but the procedure is **not** rhythmic by any definition. Each trial in the predictable condition included a constant interval as well as a jittered inter-trial interval (ITI), a varying fixation stabilization period, and varying response interval; thus, the outcome was not rhythmic. In the following analysis we demonstrate computationally that the fixed condition was not more rhythmic than the random condition.

For each block, we calculated the time interval between consecutive target onsets. This difference captures all timing related factors: ITI +gaze contingent compliance time + foreperiod + RT. For each block we computed the standard deviation of these time differences to produce a measure of rhythmicity. The standard deviation from each block was averaged according to condition (fixed/random). A paired samples t-test did not yield a significant difference (fixed: mean = 1.32, SD = 2.33, random: mean = 1.66, SD = 1.57; $t(19)=-0.53$, $p=0.7$). Moreover, a Bayesian equivalent paired samples t-test (using JASP with default parameters) resulted in a BF01 of 3.79, supporting the null hypothesis that rhythmicity is similar between conditions.

We agree that the reported effect is likely to involve an autonomic reaction, but we respectfully disagree with the belief that it has “nothing to do with expectation”. The system driving the observed effects should be able to learn the temporal meaning of the cue, i.e. to “understand” that a brief tone indicates a specific timing of the target. As the time intervals among trials vary, the cue has to be cognitively interpreted to be used for target prediction. We cannot rule out that the autonomic system is capable of this kind of complicated top-down inferences. In fact, this would have been a ground-breaking finding. However, considering that we have not used a rhythmic procedure, it is highly unlikely that the effect is driven solely by the autonomic reaction of the nervous system, without the involvement of top-down expectations mechanisms. Considering what is known on the autonomic system, it could likely be involved in reducing eye movements, but in this study, it could not do so independently, without getting temporal information from higher expectation mechanisms.

Despite our disagreement with this critique, we have now included a new experiment, in which we reduced the number of interval repetitions in the predictable condition: 80% fixed intervals and 20% of a different interval (chosen to not be a harmonic of the first). We still found the same oculomotor inhibition effect with this reduced regularity. This new experiment provides converging evidence and further clarifies that our effects are not due to rhythmicity.

We note that the experiment that you have suggested is very interesting, but unfortunately, it is not relevant to our present study because temporal attention rather than expectation would have been manipulated. For a distinction between temporal attention and temporal expectation see our recent publications: Denison, Heeger & Carrasco, *Psychonomic Bulletin Review*, 2017;

Fernández, Denison & Carrasco, *Journal of Vision*, 2019; Denison, Yuval-Greenberg & Carrasco, *Journal of Neuroscience*, 2019). Moreover, we know from previous experiments in our lab, that using similar procedures to the task that you suggested would require an explicit association between cue identity and interval, and this would be inconsistent with our goal in the present study to investigate implicit expectations.

Reviewer #3

Unfortunately, I am not able to find the answer to my concern in the rebuttal or the modified article. My concern is that the results were expected. An anticipated auditory stimulus prepares the attentional system through which it can pause other modalities and actions in order to better process the upcoming stimulus. In response to the same objection from Reviewer 2, the authors argue that since their design is not rhythmic it is not drawing an autonomic reaction of the nervous system. I would like to draw their attention to the findings of Zhao, Al-Aidroos, Turk-Browne that "Attention is spontaneously biased toward regularities." The regularity within the predictable condition is all what it takes to prepare the subjects to draw attention and I couldn't expect any result except what is shown.

I can completely understand that maybe there are crucial aspects that I am missing and other reviewers see that. I just couldn't imagine seeing any other result: an anticipated stimulus is expected to draw attention and takes the sensory resources toward its processing and put a dent in the processing of other signals and execution of motor commands. I would leave it for other reviewers and editors to decide if the paper is beyond this result or if this result was not known already.

– Thank you for your comments.

We agree that regularities modulate expectation is known from numerous studies (including the one you mentioned) and was the basic assumption of our study and the motivation of our experimental design. In fact, we open our manuscript stating, "Temporal expectations are formed by the cognitive system based on temporal regularities, and are used to distribute processing resources effectively across time." However, there is a very large gap between those studies and our current finding that very small involuntary fixational eye movements are reduced as part of this preparation, and that this reduction happens not only when expecting a visual stimulus but also when expecting an auditory one. This link between the auditory modality and the oculomotor system regarding temporal expectations is the novelty of our study and its contribution to this field.

We respectfully note that your view that this is an expected finding is not held by everyone – indeed, Reviewer 2 wrote, "there is no obvious theoretical reason to expect oculomotor inhibition prior to auditory targets". More importantly, there is no doubt that the finding is novel, as it had never been examined before. Our study contributes to the field both methodologically, as it can be used as an index for temporal expectation, and mechanistically, as a first step in understanding the interaction between modalities in expectation processes.

We agree with your comment that the effect could be driven by a crossmodal interaction between the modalities, and does not necessarily reflect a supramodal expectation mechanism, as we had suggested in the previous version. This interpretation is consistent with our view (described in the conclusion, **page 21**) that the oculomotor inhibition effect demonstrates how visual exploration pauses momentarily to allow for the processing of a predictable stimulus, regardless of its modality. Accordingly, we have modified the manuscript considerably to account for this hypothesis (e.g., **page 2 line 62; page 3 line 89**). Furthermore, we have added a section in the discussion describing this possible interpretation (**page 21, lines 544 – 561 & 575-577**). We hope that you will find this new text to be a good discussion of the potentially involved mechanisms.

Reviewers' Comments:

Reviewer #2:

Remarks to the Author:

It is unfortunate that the authors again did not take my concern serious. They decided to counteract in multiple ways. The first argument is about the exact definition of rhythmic. They claim that it is not rhythmic but instead they write "in our procedure there was a fixed interval between the cue and the target makes this a predictable or regular condition". My whole point is of course whether you call it rhythmic or not, that cue and target predictability generates "an autonomic reaction of the nervous system that has nothing to do with "expectation". This is the critical point regardless of whether you call it rhythmic or not. In their rebuttal letter it is at least acknowledge that the autonomic system could be responsible for the effect (they do not include this argument in their manuscript however). The new experiment that is added does not solve any of the concerns raised before. Finally, the experiment that I suggested is considered to be inappropriate because my experiment would have been about temporal attention and not about "temporal expectation". I read the Denison et al 2017 PBR paper that the authors refer to, which is assumed to explain the difference between temporal attention (the experiment I suggested) and the current study that is about temporal expectation. After reading this PBR paper, I am even less convinced that there is (or should be) a difference between temporal attention and temporal expectation other than that these so-called temporal expectations are nothing else than an autonomous reaction of the nervous system.

Reviewer #3:

Remarks to the Author:

By stating the limitations of the study and other potential interpretations (which somehow limits the scope of the paper) the authors have addressed my concerns.

Response to reviewers

Reviewer #2

It is unfortunate that the authors again did not take my concern serious. They decided to counteract in multiple ways. The first argument is about the exact definition of rhythmic. They claim that it is not rhythmic but instead they write “in our procedure there was a fixed interval between the cue and the target makes this a predictable or regular condition”. My whole point is of course whether you call it rhythmic or not, that cue and target predictability generates “an autonomic reaction of the nervous system that has nothing to do with “expectation”. This is the critical point regardless of whether you call it rhythmic or not. In their rebuttal letter it is at least acknowledge that the autonomic system could be responsible for the effect (they do not include this argument in their manuscript however). The new experiment that is added does not solve any of the concerns raised before. Finally, the experiment that I suggested is considered to be inappropriate because my experiment would have been about temporal attention and not about “temporal expectation”. I read the Denison et al 2017 PBR paper that the authors refer to, which is assumed to explain the difference between temporal attention (the experiment I suggested) and the current study that is about temporal expectation. After reading this PBR paper, I am even less convinced that there is (or should be) a difference between temporal attention and temporal expectation other than that these so-called temporal expectations are nothing else than an autonomous reaction of the nervous system.

With all due respect, our disagreement with the reviewer still holds. Following are our main arguments:

- (1) The reviewer agrees that our design manipulated predictability (“cue and target predictability generates an autonomic reaction of the nervous system” but s/he argues whether that this has anything to do with expectations. We think that an onset of the stimulus cannot be predictable without a system that can extract regularities, i.e. form expectations. We find oculomotor effects that are precisely timed to match the onset of an auditory target. How can this be achieved without temporal expectation?
Importantly, the experimental designs we used in both experiments 1 and 2 are commonly used in this field. Thus, the criticism raised by the reviewer would apply to numerous published studies that used a block design to study temporal expectations (e.g., Amit et al., 2019; Coull & Nobre, 1998; Elliott, 1970; Jaramillo & Zador, 2011; Klemmer, 1957; Lee, Clifford, & Arabzadeh, 2019; M.Bausenhard, Rolke, & Rolf, 2008; Mattes, 1997; Muller-Gethmann, Ulrich, & Rinkenauer, 2003; Niemi & Näätänen, 1981; Sollers J. Jhon & Hackley A., 1997).
- (2) Regarding the reviewer’s suggestion that the autonomic system mediates the observed effects, as we have acknowledged in our previous response, we don’t know which brain system is involved in this “expectation” process. Theoretically, it is not impossible that the “expectation” system would be mediated by the autonomic system, but this is extremely unlikely, as it would require that the autonomic system be able to interpret the temporal regularity of the cue in the absence of any periodicity. Critically, even under the unlikely hypothesis that the autonomic system mediates the reported effect without top-down involvement, our interpretation that oculomotor inhibition reflects an expectation process still holds. Following this comment, we now discuss this possibility in the manuscript (pages 22-23).
- (3) In the previous rounds, the reviewer’s claim of an involvement of the autonomic system was based on the suspected role of rhythmicity. This concern was disproved by the second experiment, which following the reviewer’s previous comments, was designed to be less recurring. Thus, we are puzzled by the objection that “the new experiment that was added does not solve any of the concerns raised before”. For the autonomic system to be solely responsible for the findings of the second experiment, it would have to not only interpret the predictable cue but also be able to apply statistical learning to base its influence on the most frequent intervals. And most importantly, even though this suggestion is even more unlikely, it is not inconsistent with our interpretation.

- (4) With all due respect, we chose to not include the suggested experiment in this study because, for multiple reasons, it would not address its research question: The suggested experiment requires high attentional load, top-down control and explicit instructions. In our study we never gave explicit instructions differentiating the two conditions or announced any difference between the two conditions. In contrast, the suggested experiment would require an explicit differential instruction; to present two types of tones, each signaling a different cue-target interval with 100% validity. This means that in each trial there are only two possible target timings and the cue indicates which of these two should be attended. Expectations in this case are formed in both trial types, but the cue indicates whether the earlier or the later timing is the one to be attended, manipulating temporal *attention*. In our design we manipulate temporal *expectations*: in one block there are five possible target timings and expectations are low, in another block there is only one possible timing (or one highly likely timing, in experiment 2) and expectations are high. The reviewer questions the distinction between temporal attention and expectation, without specifying his/her reasoning or citing references. This distinction in the temporal domain, which has been discussed in previous studies (Denison, Heeger, & Carrasco, 2017; Denison, Yuval-Greenberg, & Carrasco, 2019; Fernandez, Denison, & Carrasco, 2019), is based on the well-established distinction between attention and expectation in both the spatial domain (Rungratsameetaweemana, Itthipuripat, Salazar, & Serences, 2018; Rungratsameetaweemana & Serences, 2019; Summerfield & Egnér, 2009; Wyart, Nobre, & Summerfield, 2012) and the feature domain (e.g., Kok, Failing, & de Lange, 2013; Kok, Jehee, & de Lange, 2012; Kok, Rahnev, Jehee, Lau & De Lange, 2012; Summerfield & Egnér, 2016). A scientific debate regarding the existing distinction may be in place, but an anonymous review is not the right platform for such a debate.

Reviewer #3

By stating the limitations of the study and other potential interpretations (which somehow limits the scope of the paper) the authors have addressed my concerns.

Thank you.